# The Experimental Infection of Goats with Small Ruminant Morbillivirus Originated from Barbary Sheep

**DOI:** 10.3390/pathogens11090991

**Published:** 2022-08-30

**Authors:** Milovan Milovanović, Klaas Dietze, Sunitha Joseph, Ulrich Wernery, Ajith Kumar, Joerg Kinne, Nissy Georgy Patteril, Bernd Hoffmann

**Affiliations:** 1Friedrich-Loeffler-Institut, Südufer 10, 17943 Greifswald-Insel Riems, Germany; 2Central Veterinary Research Laboratory, Dubai P.O. Box 597, United Arab Emirates; 3Hatta Conservation Area, Q4W5+3JJ-Unnamed Road, Dubai P.O. Box 597, United Arab Emirates

**Keywords:** peste des petit ruminant, small ruminant morbillivirus, goats, barbary sheep, experimental infection, virulence, pathogenesis, pen-side test

## Abstract

Peste des Petits Ruminants (PPR) is a transboundary contagious disease in domestic small ruminants. Infections with the small ruminant morbillivirus (SRMV) were regularly found in wildlife, with unknown roles in PPR epidemiology. In order to access infection dynamics and virulence, we infected German Edelziege goats intranasally with a SRMV isolate that originated from Barbary sheep from an outbreak in the United Arab Emirates. Six goats were infected with cell culture-isolated SRMV, and two goats were kept in contact. Goats were daily monitored, and clinical score was recorded. EDTA blood, nasal, conjunctival and rectal swab samples were collected for the detection of SRMV genome load and serum for serological analysis. Short incubation period in infected (4 to 5 dpi) as well as in contact goats (9 dpi) was followed by typical clinical signs related to PPR. The highest viral load was detectable in conjunctival and nasal swab samples with RT-qPCR and rapid pen-side test. Specific antibodies were detected at 7 dpi in infected and 14 dpi in contact goats. In general, high virulence and easy transmission of the virus originated from wildlife in domestic goats was observed. The virus isolate belongs to Asian lineage IV, genetically related to Chinese and Mongolian strains.

## 1. Introduction

Peste des Petits Ruminants (PPR) is a transboundary contagious disease of domestic small ruminants caused by Small Ruminant Morbillivirus (SRMV), earlier known as Pest des Petits Ruminants Virus (PPRV), member of the genus *Morbillivirus* [1].

Genetically, SRMV is divided into four lineages (lineages I-III are only present in Africa and lineage IV in Africa and Asia), but only one serotype exists to date [2,3]. The high morbidity and mortality rates of PPR result in severe socioeconomic impact for sheep and goat holders. Therefore, PPR is recognised as notifiable animal disease and assigned to be eradicated by the end of 2030 by the Office International des Epizooties (OIE) and Food and Agriculture Organization (FAO) [4,5]. 

Infection in susceptible animals with SRMV is mainly intranasal through direct or indirect contact with infected animals [6,7]. In general, acute-course and serious clinical signs are characteristics for an infection with SRMV, but severity of clinical signs can vary depending on species, age, lineage and isolate [8,9,10,11]. Infection with SRMV is mainly followed by fever, apathy, anorexia, mucopurulent ocular and nasal discharge, caseous necrosis of oral and nasal mucous membranes, pneumonia and waterish to bloody diarrhoea with lethal outcome [7,12]. Experimental studies showed that goats are more susceptible to infection with SRMV and suffering from more severe clinical disease than sheep [13]. Even though, goats are described as highly susceptible for SRMV and suffering of severe disease, in not all cases the infected goats developed the severe clinical picture followed by lethal outcome. This is mainly depending of the virulence of the SRMV lineage and isolates [8,10].

Next to sheep and goats, the host range of SRMV in recent years has been expanded to a diverse range of domestic and wild species. The nomadic way of farming, the increased multispecies husbandry and the high contagiousness of SRMV all facilitate the transmission of SRMV to other animal species than just small domestic ruminants. Susceptibility to SRMV has been described in some small wild ruminants, cattle, camels, buffalos, wild boars and pigs [14]. PPR in domestic species such as cattle, buffalos and camelids pass largely unnoticed and can only be confirmed serologically [15,16,17]. Furthermore, cattle and camelid represent dead end hosts for SRMV [18]. However, in experimentally infected pigs, mild to moderate clinical signs were observed together with horizontal virus transmission to in contact goat and pig [19]. On the other hand, different wild small ruminants belonging to the subfamilies of *Caprinae* and *Antilopinae* were shown to be more susceptible to SRMV infection developing severe clinical signs with high morbidity and mortality [20,21]. There is a potential risk of virus transmission from wildlife to domestic small ruminants as intraspecies transmission was observed in Mongolian wildlife [21]. The clinical course of disease in small wild ruminants is poorly understood, as reported field observations only capture data at the late stage of a SRMV infection and experimental data are scarce. In most cases, high mortality and characteristic pathomorphological findings were the main links for suspicion of SRMV infection, followed by laboratory confirmation [20,21]. Many PPR-affected countries were already reporting PPR in wildlife, indicating that SRMV spill-over events are not rare [14]. Difficulties in controlling the interface of wildlife and small ruminants, in particular in pastoral settings, could make successful eradication challenging when the epidemiological role of wildlife as disease reservoir remains unclear.

In this study, we performed the molecular characterization of a SRMV isolate from an outbreak in wild Barbary sheep (*Ammotragus larvia*) in the United Arab Emirates (UAE). Furthermore, we analysed this SRMV isolate in an experimental infection in vivo to understand the pathogenesis in and the transmission potential to domestic goats. Here, the dynamic of the infection was followed by virological and serological methods including the easy-to-use rapid pen-side tests. Our results provide important insights into the potential of SRMV isolates from wild animals to infect domestic goats and its diagnostic detection.

## 2. Results

### 2.1. Outbreak Investigation 

Main clinical signs observed in the field in affected animals were mucopurulent nasal discharge, multiple necrotic erosions of the mouth mucosa, mainly located on the soft palate and gums, diarrhoea, and in the most of the pregnant Barbary sheep, abortion. According to field observation, total morbidity rate was about 80%, with a mortality rate of 65–70% in Barbary sheep and mortality rate of 1–2% in other ungulates, especially in Scimitar oryxes. Collected tissue samples had genomic load estimated by RT-qPCR between Ct 18 to 25.

### 2.2. Virus Isolation 

SRMV characteristic cytopathogenic effect (CPE) in a form of syncytial formation with destruction of cell monolayer was observed starting in the second passage after 5 days of incubation and progressed to cell rounding and aggregation on day 7. Successful SRMV isolation was confirmed by genomic load estimated by RT-qPCR (original material Ct 18.8 and second passage Ct 19.4). Cell culture flask containing VDS-SRMV suspension was freeze/thawed one time at −80 °C for 24 h. After thawing material was aliquoted and 10^4,0^TCID_50_/_mL_ SRMV titer was determined. Rest of the material was stored at −80 °C until further use in animal experiment.

### 2.3. Experimental Infection of Goats

#### 2.3.1. Clinical Observation

Clinical signs in SRMV infected goats started on 4 days post infection (dpi) with mild fever in four goats (Z/9, Z/10, Z/11 and Z/23), and 5 dpi in two remaining inoculated goats (Z/20 and Z/22) followed by the in-contact goats with body temperature rise on 8 dpi (Z/25) and 10 dpi (Z/24). In addition, one of the in-contact goats (Z/25) had watery oculo-nasal discharge on 9 and 10 dpi with normal body temperature. Majority of infected goats had the peak of body temperature at 8 dpi (Z/10–41.3 °C, Z/11–41.5 °C, Z/20–41.1 °C and Z/23–39.9 °C), other two infected goats reached peak at 7 dpi (Z/9–40.7 °C and Z/22–40.6 °C) and the in-contact goats peaked on 10 dpi (Z/24–41.0 °C) and 13 dpi (Z/25–40.6 °C). Then body temperature slowly dropped until euthanasia of the goats. Detailed daily body temperature is presented in Figure 1. 

High body temperature was followed by depression, oculo-nasal discharge, necrotic lesions of facial mucosa, diarrhoea and respiratory signs. Severe clinical signs occurred between 5 dpi to 12 dpi in infected goats and between 10 dpi to 17 dpi in in-contact goats.

General condition observation revealed a slight inactivity in all infected goats at 6 dpi and in in-contact goats at 10 dpi (Z/24) and 11 dpi (Z/25). Mild inappetence was observed in two goats at 7 dpi (Z/20) and 8 dpi (Z/11) and inactivity, apathy and anorexia in goat Z/9 at 11 dpi. Depression, inability to stand, lethargy and dehydration was found in five goats at 10 dpi (Z/23), 12 dpi (Z/10 and Z/22) and 17 dpi (Z/24 and Z/25). Watery oculo-nasal discharge was observed starting from 5 dpi in two goats (Z/11 and Z/23), 6 dpi in three goats (Z/9, Z/20 and Z/22), at 7 dpi in goat Z/10, at 9 dpi in goat Z/25 and at 11 dpi in goat Z/24. Mucoid oculo-nasal discharge with mild conjunctivitis were observed in goat Z/20 at 8 dpi, whereas mucopurulent oculo-nasal discharge with conjunctivitis observed in six goats at 9 dpi (Z/23), 10 dpi (Z/9 and Z/11), 12 dpi (Z/10), 16 dpi (Z/24) and 17 dpi (Z/25). Goat Z/22 developed at 12 dpi mucopurulent nasal discharge and severe conjunctivitis with profuse mucopurulent ocular discharge (Figure 2A). Congestion of oronasal mucosa was seen in infected goats at 5 dpi (Z/23), 6 dpi (Z/11 and Z/22), and 7 dpi (Z/9, Z/10 and Z/20) and in one in-contact goat at 14 dpi (Z/25). All goats developed different intensity levels of lesions such as pin-prick lesions in buccal cavity seen in two infected goats at 8 dpi (Z/20) and 9 dpi (Z/11), clear erosive lesions on oronasal mucosa were observed in one infected goats at 10 dpi (Z/9), and in in-contact goats at 15 dpi (Z/24) and at 16 dpi (Z/25). Erosive/ulcerative lesions in buccal cavity, nasal mucosa and nares with oedematous lips observed in three infected goats at 10 dpi (Z/22), 11 dpi (Z/11) and 12 dpi (Z/10) (Figure 2B). Soft faeces were noticed in infected goats at 5 dpi (Z/11), 6 dpi (Z/23), 7 dpi (Z/9 and Z/10), and in in-contact goats at 10 dpi (Z/24) and 12 dpi (Z/25). The change from normal to runny faeces was noticed at 7 dpi in goat Z/20 and in goat Z/22 at 8 dpi. In four goats frank diarrhoea was noticed in two infected goats at 10 dpi (Z/9 and Z/11) and both in-contact goats at 15 dpi (Z/24 and Z/25), whereas in the other four infected goats haemorrhagic diarrhoea at 8 dpi (Z/20), 10 dpi (Z/23) and 12 dpi (Z/10 and Z/22) was observed (Figure 2C). Respiratory signs were less severe and they weren’t observed in one infected (Z/20) and one in-contact goat (Z/25) throughout all experiment. Slight tachypnoea was noticed in infected goats at 7 dpi (Z/11), 8 dpi (Z/10 and Z/23), 9 dpi (Z/9), 10 dpi (Z/22) and in one in-contact goat at 14 dpi (Z/24). Severity of observed respiratory signs ranged from mild cough in three goats at 10 dpi (Z/23 and Z/10) and 12 dpi (Z/9), via tachypnoea and dyspnoea with present constant coughing at 12 dpi in goat Z/22. Prior to euthanasia of infected goats loss of weight together with dehydration, due to progressive watery diarrhoea and rejection of food and water uptake was observed (Figure 2D). Goats were euthanized due to ethical reasons starting from 8 dpi (Z/20), 10 dpi (Z/23), 12 dpi (Z/9, Z/10, and Z/22), and 17 dpi (Z/24 and Z/25). From all goats in the study, only one infected goat (Z/11) recovered, starting with recovery of the body temperature (11 dpi), followed by recovery of respiratory signs (14 dpi), mucosal lesions (17 dpi), general condition (18 dpi), oculo-nasal discharge (23 dpi) and diarrhoea (26 dpi). The clinical score is summarized in Figure 3.

#### 2.3.2. Gros Pathology

Conjunctivitis with dilatation of scleral blood vessels was noted (Figure 2E). Cyanosis and caseous necrotic lesions were observed on the mucous membranes of the oral and nasal cavities. In the oral cavity, multifocal caseous necrotic lesions were localized mainly on the dental pad, the inner surface of the upper lip, the anterior and posterior gums of the mandibular incisors, and the dorsal surface of the tongue (Figure 2F). In the nasal cavity necrotic lesions were localized on the lateral sides and septum nasi of the anterior and middle part of the nasal cavity (Figure 2G). Only the mesenteric lymph nodes were enlarged. In the lungs, edema and diffuse mottling with red and gray surfaces were seen. Pathomorphological changes in the liver and spleen were not observed. Pathomorphological investigations of other organs belonging to digestive, respiratory or urogenital tract were not carried out.

### 2.4. Serological Analysis

Detection of SRMV specific antibodies with cELISA in infected goats was possible starting from 7 dpi in 4 goats (Z/9, Z/10, Z/11 and Z/23) with clear positive results; one goat had questionable results (Z/22), and one goat was detected as negative (Z/22). Except one euthanized goat (Z/20), all infected goats had strong antibody responses detected at 10 dpi. Both in-contact goats had strong antibody respond detected at 14 dpi. Detailed results are presented in Figure 4.

### 2.5. Virus Detection

#### 2.5.1. ID Rapid PPR Antigen Dipstick Field Test

Analysing nose swab samples, wild first positive results were obtained at 5 dpi in infected goats ranging from questionable (Z/22), low positive (Z/11 and Z/20), moderate positive (Z/9 and Z/23) to high positive (Z/10). More uniform results were obtained at 7 dpi with detection of all infected goats as high positive. First questionable positive results in one in-contact goat (Z/24) was obtained at 10 dpi. At 12 dpi in-contact goats were detected as low positive (Z/25) and high positive (Z/24). All goats were detected as high positive until they were euthanized, except recovered goat (Z/11), which was detected positive up to 17 dpi. Detailed results of ID rapid PPR antigen field test from nasal swab can be found in Table 1 and Appendix A.

First positive conjunctival swab samples were detected at 5 dpi in three infected goats as low positive (Z/9, Z/10 and Z/22) and one as high positive (Z/23). At 7 dpi all infected goats were detected positive ranging from low positive (Z/20), moderate positive (Z/23) and high positive (Z/9, Z/10, Z/11 and Z/22). Both in-contact goats were detected as questionable at 10 dpi. The same was observed in results from nasal swab:all goats have been detected as high positive until they were euthanized, except recovered goat (Z/11) which was detected positive up to 14 dpi. Detailed results of ID rapid PPR antigen field test from conjunctival swab can be found in Table 2 and the Appendix A.

#### 2.5.2. Molecular Analysis

##### Reverse Transcriptase Real-Time Polymerase Chain Reaction (RT-qPCR) Results of Swab and EDTA Blood Sample Material

The first detection of SRMV genome at 3 dpi was in the nasal swabs of infected goats Z/22 and Z/23, and in the EDTA blood of infected goats Z/10 and Z/23. At 7 dpi, both in-contact goats (Z/24 and Z/25) became RNA positive in nasal and conjunctival swabs, as well as the EDTA blood of one in-contact goat Z/24. An increase in the viral genome load of all inoculated goats was observed until 7 dpi in the EDTA blood samples and until 10 dpi in all swab samples. Highest viral loads were observed in conjunctival swabs (Z/9, Z/10, Z/23, Z/24 and Z/25), nasal swabs (Z/11, Z/20 and Z/22) and rectal swab samples. In summary, swab material showed a higher genome load then the EDTA blood samples of the respective animals. An increase of the viral genome load in the contact goats was observed in swab sample material until 17 dpi and in EDTA blood sample until 14 dpi. All RT-qPCR results are shown in Figure 5.

##### Organs

The detection of the SRMV genome was possible in all tissue sample material except from the liver of the recovered goat (Z/11). The results of SRMV genome load in tissue sample material are presented in Figure 6.

##### Phylogenetic Analysis

The complete genome sequence obtained in this study from Barbary sheep has been submitted to National Centre for Biotechnology Information (NCBI) under accession number OM867572. The BLAST results of the full-length nucleotide sequences showed that the SRMV strain from Barbary sheep has the highest nucleotide similarity of 98.6% (Query coverage of 100%) with the SRMV isolate Sungri 1996 MSD (KJ867542) and with the SRMV isolate Izatnagar/94 (KR140086). Phylogenetic analysis including 24 SRMV full-genome sequences from NCBI with already assigned lineages revealed that the SRMV strain from Barbary sheep fell into lineage IV and in the same cluster as SRMV isolates from Chines/Mongolian region (shown in Figure 7). 

## 3. Discussion

The continued spread of the SRMV host range, especially in small wild ruminants, raises concerns about the successful control and eradication of PPR worldwide. Wild ruminants can play an important role as a potential virus reservoir and pose a threat for the local spread of the virus to domestic small ruminants and new areas. At the moment infection dynamics and virulence of SRMV strains from wildlife is poorly understood. In order to fill some knowledge gaps in this regard, the present study followed the disease course in domestic goats after infection with a SRMV strain isolated from wild Barbary sheep.

In experimental studies, clinical signs ranging from mild [11] to severe causing high case fatality in infected goats [7,8,22] have been described. In our study, using the intranasal route of infection, the high virulence of the virus was confirmed by a severe clinical course in all goats, like it was described in the field. Severity of the clinical signs is mainly related to species, breed, SRMV strain, and for experimental settings the number of passages of SRMV in cell culture and virus storage conditions [7,8,10,11]. In order to best mimic a natural infection, we used young goats as the most susceptible sub-population and internasal application for infection. Furthermore, the cell culture isolated SRMV strain had a minimal passage history and was stored at −80 °C until usage in the animal trial. Altogether, the SRMV isolate from Barbary sheep produced a high infection rate and could easily transmitted to in-contact goats. All animals developed typical clinical signs for PPR observed in other experimental studies and in the field.

In our study, incubation period in infected goats was between 4 dpi to 5 dpi. The first observed clinical sign was high body temperature followed by diverse PPR related clinical signs like general condition disorder, oculo-nasal discharge, necrotic lesions in oronasal mucosa, diarrhoea and respiratory signs. The in-contact goats developed of the same clinical signs beginning at 9 dpi of the experimentally infected animals suggested they picked up the infection right at the onset of disease in that group. Peaks of clinical signs in infected goats were observed between 9 dpi and 12 dpi and in the contact goats at 17 dpi. Short incubation period and the expression of different intensity of clinical signs observed in this study is in the line with observations from other experimental studies using goats and high virulent SRMV isolates [7,8,9,12]. Even though the isolate used in this study shows to be highly virulent causing severe clinical signs in infected and in-contact goats, only one infected goat recovered. This recovery from the infection with high virulent SRMV strain has been addressed to multi factors, such as general condition of the infected animal, avoid secondary bacterial infection specifically with *Mannheimia haemolytica* responsible for bronchopneumonia and immunosuppression effect of the virus [7,8,11,23,24]. The recovered goat (Z/11) developed milder depression and inappetence together with slight tachypnoea compared to the other goats which can be addressed to good general condition and absence of secondary bacterial infection.

Pathomorphological lesions observed in this study were mainly located in the gastrointestinal and respiratory tract embracing regional lymph nodes. The intensity of the pathomorphological lesions of SRMV to this organic system mainly depends on the virulence of the used strain, as the mild changes were seen with mild strain and severe changes with high virulent SRMV strains used [7,11,22,25].

Seroconversion was observed in infected goats from 7 dpi and in in-contact goats from 14 dpi. In a previous experimental study, it has been shown that early antibody response didn’t influence the severity of the clinical signs [8]. In the same study it was noticed that some infected animals with a strong antibody response recovered from the infection. In our study the surviving goat (Z/11) showed a humoral response very comparable to the other inoculated goats. Therefore, based only on the level of seroconversion, the recovery of the goat Z/11 starting at 12 dpi cannot be explained.

After internasal infection and during incubation period, initial replication of the SRMV occurs in regional lymph organs of head which is followed by further dissemination of the virus to distanced lymphatic organs by peripheral blood leukocytes (PBL) [9,11]. Many authors confirm this finding as detection of the virus genome during incubation period was possible from conjunctival, nasal or mouth swab samples [10,11,26]. These findings were also confirmed in our study as the detection of the SRMV at 3 dpi in nasal swab and at 5 dpi in conjunctival swab was successful by RT-qPCR and at 5 dpi in nasal and conjunctival swab by ID rapid PPR antigen field test. In the in-contact goats the presence of SRMV was detected at 7 dpi by PCR and at 10 dpi by ID rapid PPR antigen field test in nasal and conjunctival swab samples, respectively. The lower sensitivity of ID rapid PPR antigen field test compared to the RT-qPCR was in line with previous published results and the possibility of RT-qPCR to multiply targeted genome sequences [27] It was described that nasal swab samples can provide unambiguously results compare to PCR, but despite lower sensitivity of ID rapid PPR antigen field test, the detection of false positive samples was not observed in both types of matrices [27]. 

From RT-qPCR results of all goats included in the study, higher genome load was observed in swab samples compare to EDTA blood, which is also in line with published results [11,24,25]. In our study the highest genome load was observed in conjunctival swab samples, followed by nasal and rectal swab samples. The conjunctival swab as the sample with highest SRMV genome load was already observed in other experimental studies [22,25]. 

A slightly lower genome load was observed in rectal swabs. Previous studies showed that successful SRMV isolation was possible from the rectal swab in the first days of infection and the antigen can be detected over the long period of time (up to 12 weeks) in recovered animals [28,29]. For the non-invasive SRMV monitoring in wildlife the use of faecal samples was described. Here, the investigation of samples as fresh as possible is recommended [30].

In an experimental study investigating the length of SRMV excretion from infected animals, it has been described to be not longer than 26 dpi in nasal, mouth and ocular swab samples [31]. In our study, one goat (Z/11) survived the infection and could be tested over the time until the end of the experimental study at 28 dpi. All swab and EDTA blood samples collected from this goat in the late phase of disease curve were positive by RT-qPCR, but negative by the ID rapid PPR antigen field test. This difference can be explained by the strong immunological response/high antibodies titre in late stage of the infection, most likely responsible for blocking the antigen detection by the ID rapid PPR antigen field test [8,9,11]. This statement was confirmed by detection possibility of live and antibody neutralized SRMV cell culture isolate using ID rapid PPR antigen field test, which provided clear positive results of live virus and questionable result of antibody neutralized virus (Appendix A). Viability and virus load of live as well as antibody neutralization SRMV cell culture isolate was analysed by virus isolation on VDS and genomic load estimated by RT-qPCR (live virus Ct 21.64 and antibody neutralized Ct 22.53), respectively.

The detection of SRMV genome in all collected organ samples was feasible from all goats except in the liver of the recovered goat Z/11. Previous studies showed that high genome load was found in the lymphatic tissue and in lungs which is in with the lymph- and epitheliotropic nature of the SRMV [11,22,32]. This is reflected in our results as the euthanized goats had over all high genome load in lymph nodes and spleen, and in the lungs of recovered goat, but no virus detection in the liver. Consequently, in the case of fatality, when infection of PPR is suspected, the sample material of choice should be lymphatic tissue or lungs [32].

According to the global distribution of different SRMV lineages, in the Middle East, lineages III and IV are present [2]. SRMV isolates belonging phylogenetically to lineage IV can be divided into African and Asian group, with the further division of Asian isolates into two subgroups, Indian vaccine strain and Chinese strains [33]. Phylogenetical analysis of our SRMV strain from Barbary sheep revealed that the isolate belongs to the Asian lineage IV, subgroup Chinese strains. It clusters together with the SRMV strain isolated from the Chinese/Mongolian region. SRMV infection in wildlife in Mongolia was described with high mortality in affected animals, similar to the reports from the outbreak in Barbary sheep in UAE and the high infection rate observed in our experiment [34]. Nevertheless, the way of introduction of the identified SMRV strain from the Far East region to the UAE remains unclear. Several anthropogenic factors linked globalization and the trade of live animals or products represent likely pathways additionally facilitated by the high contagiousness of the virus.

## 4. Materials and Methods

### 4.1. Outbreak Investigation

An outbreak of SRMV occurred in January 2021 amongst different wild ungulate species: Barbary sheep (*Ammotragus lervia*), Arabian oryx (*Oryx leucoryx*), Scimitar oryx (*Oryx dammah*), Nubian ibex (*Capra nubiana*), Black buck (*Antilope cervicapra*), Sand gazelle (*Gazella marica*), Dorcas gazelle (*Gazella dorcas*), Mountain gazelle (*Gazella gazelle*), and Arabian gazelle (*Gazella arabica*) at a wildlife conservation area in Dubai, United Arab Emirates. As the most affected species with morbidity of 80%, Barbary sheep are also the most represented species in the area with a population of 4100 animals from a total of 4650 animals. SRMV infection was confirmed at Central Veterinary Research Laboratory (CVRL) in Dubai, UAE by commercially available ELISA kit ID Screen PPR Antigen Capture (ID, Montpellier, France) according to manufacture instruction and by reverse-transcriptase real-time PCR (RT-qPCR). Extraction of nucleic acid and amplification of the conserved region of nucleocapsid protein (Np) by RT-qPCR was done according to already published method by Batten et al., 2011 [35] using tissue samples (tongue, lung and mesenteric lymph node).

### 4.2. Virus Isolation

The same tissue sample material that originated from the Barbary sheep that were positive to SRMV were sent at ambient temperature from CVRL (Dubai, UAE) to the German National Referent laboratory for PPR at the Friedrich-Loeffler-Institute (FLI). Tissue samples were cut in lentil size pieces using scissors and tweezers and further homogenized in serum-free cell culture medium (FLI intern medium number ZB5) with double standard concentration of antibiotics and antimycotics (20,000 g/mL Penicillin, 20,000 units/mL Streptomycin, 10 mg/mL Gentamicin, 250 g/mL Amphotericin B (Gibco, Grand Island, NE, United States)) and with steal balls using the TissuLyser II tissue homogenizer (QIAGEN, Hilden, Germany) for 1 min and 20 Hz. After tissue homogenisation, samples were cooled on ice for one minute to prevent virus inactivation, then centrifuged (8000 revolution per minute (RPM), 10 s). The supernatant was used for virus isolation experiments and for RNA extraction and analysis by RT-qPCR.

One-day-old Vero cells expressing dog signalling lymphocyte activation molecule (SLAM) on their surface (FLI cell culture collection number RIE1280/57) of 90% confluency in T25 cm2 tissue culture flask with cell culture medium containing 10% of foetal calf serum (FCS) (FLI intern foetal calf serum number P85) and 10 mg/mL Zeozin (InvivoGen, Toulouse, France) were used for initial virus isolation. Then, culture medium was removed and 1 mL of supernatant from homogenized tissue material was added. Adsorption of the virus was allowed for 2 h in an CO_2_ incubator on 37 °C and 5% CO_2_ with constant slow shaking using tilth shaker. After adsorption of the virus, inoculated material was removed and cells were two times washed with serum-free ZB5 with double standard concentration of antibiotics and antimycotic. New fresh ZB5 containing 10% FCS, 10 mg/mL Zeozin and double concentration of antibiotics and antimycotic was added after washing and incubation was continued for 7 days at 37 °C and 5% CO_2_. Cells were daily monitored using stereomicroscope for potential bacterial or fungal contamination and appearance of virus cytopathogenic effect (CPE) until day 7 after which cell passage was done. 

The virus titer was determined by using a log-10 dilution series of the virus in twelve replicates using 96 well microtiter flat bottom plates on VDS cells. Reading of the plates was done after incubation of 7 days in the CO_2_ incubator at 37 °C and with 5% CO_2_. Virus titer was calculated using Spearmen and Kaerber method [36]. 

### 4.3. Experimental Design

#### 4.3.1. Experimental Inoculation of Goats

For the experimental infection, eight male German Edelziege goats of approximately six months of age were housed in the high security containment facility at the FLI, Island of Riems. The goats were acclimatized to the environment 11 days priori infection to reduce stress and get familiar to the environment. Goats originated from a goat farm in Germany and tested free from SRMV genome and antibodies priory experimental infection. All animals were healthy and in a good condition.

Six goats were intranasally infected with 2 mL of SRMV isolate from VDS cells with an infectious titer of 10^3,5^TCID_50_/mL using MAD NasalTM intranasal mucosal atomization device (Teleflex Medical, Morrisville, NC, USA). Before infection, two goats (Z/24 and Z/25) were physically separated by placing them into another room without direct contact with the rest of the goats. One day after infection, goats (Z/24 and Z/25) were placed back in the group and used for investigation of horizontal virus transmission.

Goats were daily monitored during the whole length of the experiment and the clinical score was recorded [11] to assess clinical progression of the disease and to determine proper ethical time point for euthanizing severe affected animals.

#### 4.3.2. Sample Collection 

Sample material (EDTA blood, serum, nasal, conjunctival and rectal swab) was collected before infection of the goats (0 dpi) and post infection (3, 5, 7, 10, 12, 14, 17, 21, 24 and 28 dpi). Collection of serum and EDTA blood was done using an adapter system (Kabe Labortechnik GmbH, Germany) from the jugular vein. Nasal and conjunctival swabs were collected with swab material from the ID Rapid PPR Antigen kit (ID, Montpellier, France) and immediately immersed into dilution buffer from ID Rapid PPR Antigen kit (ID, Montpellier, France). Rectal swabs were collected with cotton coated swabs (Copan Italia, Brescia, Italy) and immediately after collection immersed into 2 mL of serum-free ZB5 with double concertation of antibiotics and antimycotics. 

Euthanized goats were pathomorphological investigated, and tissue samples (lung, spleen, liver, mesenteric lymph node, and mediastinal lymph node) were collected. All sample material after collection were aliquoted and storage on −80 °C until analysis.

### 4.4. Serological Analysis

Commercially available ID Screen PPR Competition ELISA kit (ID, Montpellier, France) was used for the detection of SRMV specific antibodies in serum samples according to manufacture instructions.

### 4.5. Virus Detection 

#### 4.5.1. Pen-Side Test for SRMV Antigen Detection

Commercially available ID Rapid Peste des Petits Ruminants Antigen dipstick field test (ID, Montpellier, France) was used for the detection of SRMV antigen from nasal and conjunctival swab samples according to manufacture instructions. 

#### 4.5.2. Molecular Analyses

##### Reverse Transcriptase Real-Time Polymerase Chain Reaction (RT-qPCR) Method

NucleoMag VET kit (Macherey-Nagel, Düren, Germany) was used for the extraction of viral nucleic acid from sample material with the half-automated King Fisher platform (King-Fisher Flex magnetic particle processor, Thermo Fisher Scientific, Waltham, MA, USA). For the amplification of the conserved region of nucleocapsid protein (Np), a specific primer probe mix [37]using FAM channel (concentration of forward and reverse primer: 15 µM; probe 5 µM). As a control of extraction and amplification process, a heterologous control system [38]was implemented and co-detected in all RT-qPCR runs using the HEX channel (concentration of forward primer, reverse primer and probe: 5 µM). 

Commercially available, AgPath-ID™ One-Step RT-PCR Reagents of Applied Biosystems™ (Waltham, MA, USA) was used for the real-time RT-PCR. Briefly, 2.5 µL of template was added to the reaction mix of 1.25 µL RNase-free water, 6.25 µL 2x RT-PCR Buffer, 0.5 µL 25x RT-PCR Enzyme Mix, 1.0 µL specific Np SRMV primer probe mix and 1.0 µL EGFP mix primer probe mix and the RT-qPCR was run on the CFX 96 real-time PCR cycler (Bio-Rad, Hercules, CA, USA).

In every run, a generated SRMV standard series produced by droplet PCR (QX200 Droplet Digital PCR System, Bio Rad, Hercules, CA, USA) was included for calculating the genome copy numbers.

##### Phylogenetic Analysis

Using HTS-SISPA technology [39] and the previously described procedure [40] double sense (ds) cDNA of SRMV was prepared and submitted to Eurofins Genomic (Ebersberg, Germany) for genome sequencing on Illumina platform. Obtained fastq raw data was further processed using the Genius Prime v2021.0.1 (Biomatters Ltd., Auckland, New Zeeland) to construct complete genome sequence of the SRMV isolate. The complete genome sequence obtained in this study from Barbary sheep has been submitted to the National Centre for Biotechnology Information (NCBI) under accession number OM867572.

Mega X [41] software was used for evolutionary analysis and for creating phylogenetic trees. We used the neighbour-joining tree-built method, the genetic distinction model Tamura-Nei and bootstrap analysis with 1000 replicates. To identify the nearest molecular neighbors, the consensus sequence was blasted against the nt database on NCBI. 

## 5. Conclusions

In this study, we report for the first-time isolation of SRMV originated from the Barbary sheep. Findings in this study confirm the high virulence of used SRMV strain in domestic goats. The experimentally infected goats developed severe clinical signs and virus was easily transmitted to in-contact goats. Depending of the phase of the disease sample material as well as diagnostic method should be considered as in the early stage swab samples should be preferred over the EDTA blood and in the case of fatality lung or lymphatic tissue should be considered. Regarding the diagnostic method for antigen detection, the pen-side test can provide rapid and reliable results in the field. However, the diagnostic window of the pen-side test is limited compared to PCR and this mainly depends on the viral load and the level of specific SRMV antibodies in the sample. Especially in animals that already have SMRV antibodies, RT-qPCR should be used to reliably exclude acute SMRV infection. Early detection of high SRMV genome load in secretions and excretions, before clinical signs were detectable, indicates a high risk of early unnoticed SRMV transmission amongst small ruminants. In the pastoral and nomadic system that prevail in many PPR affected regions, including Arabian Peninsula, the sharing of grazing areas enables potential SRMV spill over between wildlife and domestic small ruminants. However, addressing SRMV in wildlife populations in order to fulfil control and eradication plans is much more challenging due to the uncontrolled animal movements and the lack of suitable vaccines. Regions with substantial SRMV susceptible wildlife populations should therefore reduce risks of SRMV spill-over into wildlife to protect wild and domestic animas alike.

## Figures and Tables

**Figure 1 pathogens-11-00991-f001:**
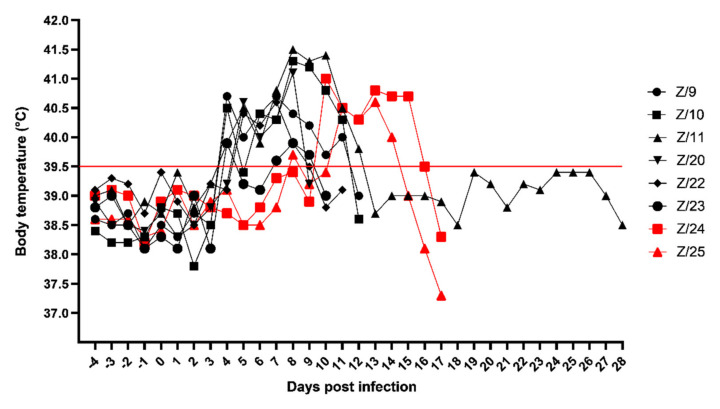
Daily body temperature of infected (black symbols) and in-contact goats (red symbols). Red line cut off, under normal body temperature, above high body temperature.

**Figure 2 pathogens-11-00991-f002:**
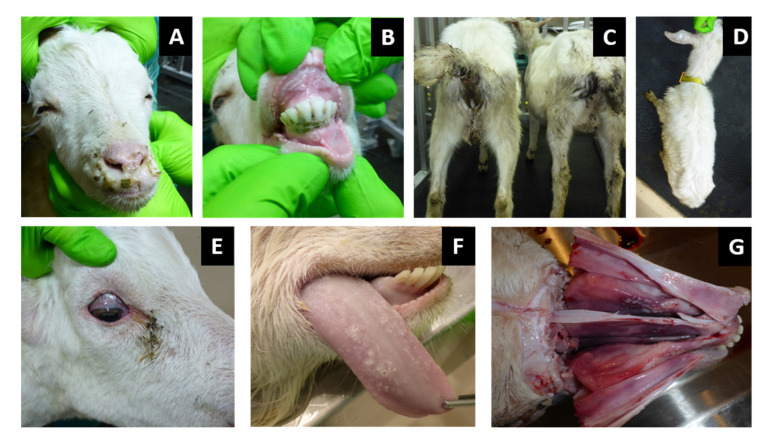
Clinical observation in infected and in-contact goats: (**A**) mucopurulent nasal discharge; (**B**) erosive/ulcerative lesions of dental pad, inner surface of the upper lip and gums of the mandibular incisors; (**C**) faecal soiled tail and anal region as a result of frank diarrhoea; (**D**) emaciated goat as a result of infection; (**E**) dried copious eye discharge with conjunctivitis. Pathomorphological observation in infected and in-contact goats: (**F**) multifocal caseous necrotic lesions on dorsal surface of the tongue; (**G**) cyanosis and necrotic lesions of anterior and middle part of lateral slides and septum nasi.

**Figure 3 pathogens-11-00991-f003:**
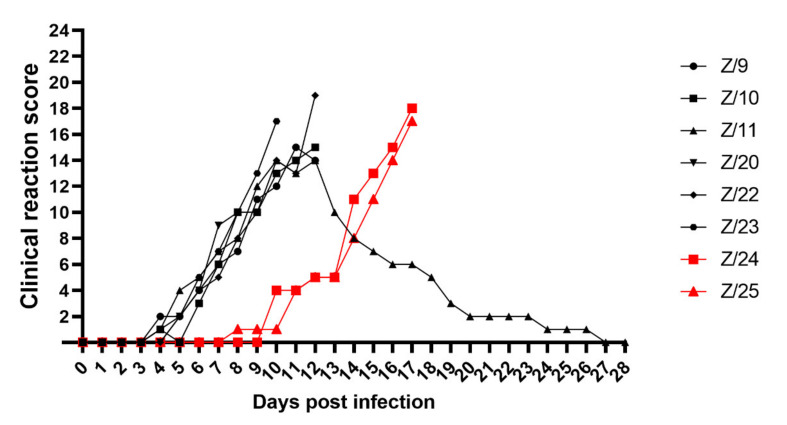
Summarized daily clinical score of infected (black symbols) and in-contact goats (red symbols).

**Figure 4 pathogens-11-00991-f004:**
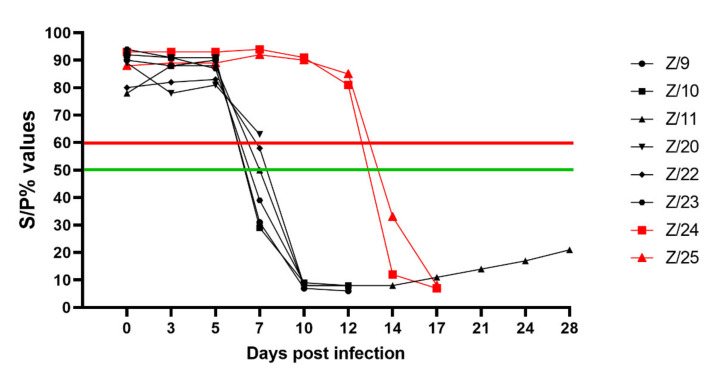
CELISA results as percentage of inhibition of infected (black symbols) and in-contact goats (red symbols). Values ≤ 50% (under green line) positive (four infected at 7 dpi, at 10 dpi five infected and both in-contact at 14 dpi), >50% but ≤60% (between green and red line) questionable (one infected (Z/22) at 7 dpi) and >60% (over red line) negative (all infected goats until 5 dpi, one infected (Z/20) at 7 dpi and both in-contact goats until 12 dpi).

**Figure 5 pathogens-11-00991-f005:**
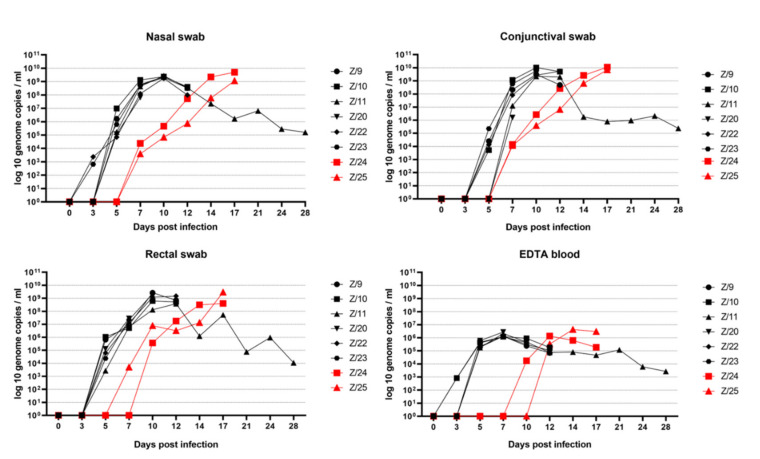
RT-qPCR results with SRMV genome load in nasal swab, conjunctival swab, rectal swab and EDTA blood over the time in infected (black symbols) and in in-contact goats (red symbols).

**Figure 6 pathogens-11-00991-f006:**
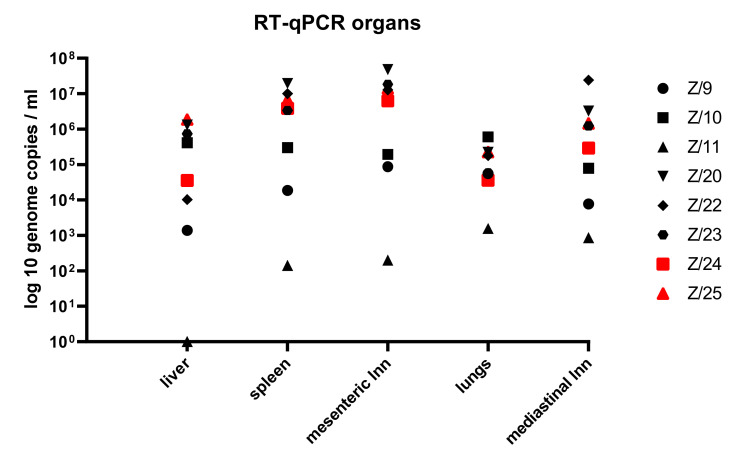
RT-qPCR results with SRMV genome load in organ material from infected (black symbols) and in in-contact goats (red symbols).

**Figure 7 pathogens-11-00991-f007:**
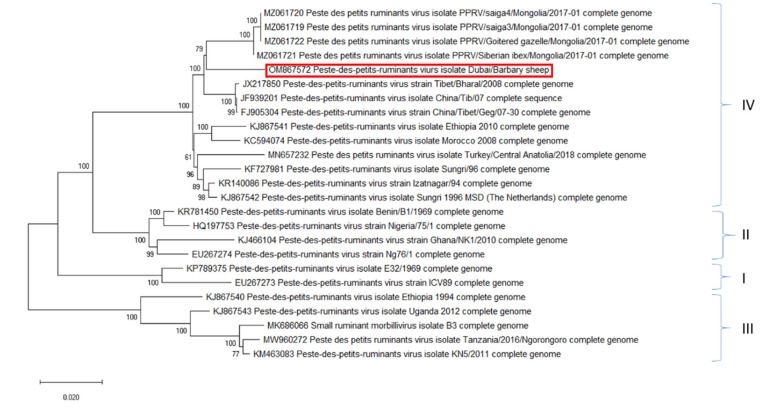
Phylogenetic analysis of the SRMV Barbary sheep genome. The phylogenetic three was created with MegaX with the statistical method Neighbour-Joining using Tamura-Nei genetic distance method and bootstrap analysis with 1000 replicates including complete genome sequences of SRMV strains representing the known lineages. Sequence in the red boxrepresents the SRMV Barbary sheep strain used in this study.

**Table 1 pathogens-11-00991-t001:** ID rapid PPR antigen dipstick field test results of nasal swab. Key: negative (−) control band visible test band not, questionable ((+)) control band visible and barely visible test band, low positive (+) test band visible but not strong colour as control band, moderate positive (++) test band same visible colour as control band, high positive (+++) test band stronger colour than control band.

ID.	Days Post Infection										
	0	3	5	7	10	12	14	17	21	24	28
Z/9	−	−	++	+++	+++	+++	^*^				
Z/10	−	−	+++	+++	+++	+++					
Z/11	−	−	+	+++	+++	+++	++	(+)	−	−	−
Z/20	−	−	+	+++							
Z/22	−	−	(+)	+++	+++	+++					
Z/23	−	−	++	+++	+++						
Z/24	−	−	−	−	(+)	+++	+++	+++			
Z/25	−	−	−	−	−	+	+++	+++			

* Gray fields—goats were euthanized and samples were not available for analyses.

**Table 2 pathogens-11-00991-t002:** ID rapid PPR antigen dipstick field test results of conjunctival swab. Key: negative (−) control band visible test band not, questionable ((+)) control band visible and barely visible test band, low positive (+) test band visible but not strong colour as control band, moderate positive (++) test band same visible colour as control band, high positive (+++) test band stronger colour than control band.

ID	Days Post Infection										
	0	3	5	7	10	12	14	17	21	24	28
Z/9	−	−	+	+++	+++	+++	^*^				
Z/10	−	−	+	+++	+++	+++					
Z/11	−	−	−	+++	+++	+++	(+)	−	−	−	−
Z/20	−	−	−	+							
Z/22	−	−	+	+++	+++	+++					
Z/23	−	−	+++	++	+++						
Z/24	−	−	−	−	(+)	+++	+++	+++			
Z/25	−	−	−	−	(+)	++	+++	+++			

* Gray fields—goats were euthanized and samples were not available for analyses.

## Data Availability

The data presented in this study are available on request from the corresponding author.

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
