# Peer review of "The Experimental Infection of Goats with Small Ruminant Morbillivirus Originated from Barbary Sheep"

_pathogens, 2022, doi:10.3390/pathogens11090991_

Round 1

Reviewer 1 Report

In this manuscript, the authors show that the PPRV strain isolated from Barbary sheep are highly pathogenic to goats. The authors also compared the method by using ID rapid PPR antigen dipstick with cELISA and qRT-PCR and provided the information on suitable specimens.

PPRV is highly pathogenic to goats and sheep and has a significant impact on these livestock. Due to significant economic losses, PPRV eradication programs have been implemented by the OIE and FAO against PPRV. Outbreaks have been observed in wild small ruminants, and clarifying the role of these animals as a source of PPRV infection is important to advance PPRV eradication.

I consider this manuscript adequately presents the results of the research on the experimental infection with the isolated strain.

I will only point out the minor points in this manuscript.

1.     Line 221. EDTA blood is negative on day7 Z/25 in Figure 6. Is there a mistake in the text?

2.    The information of the some references is missing. The journal names or the page numbers are not listed in some references.

Reviewer 2 Report

The manuscript "Experimental infection of goats with Small Ruminant Morbillivirus originated from Barbary sheep" describes the pathogenesis of SRMV isolated from Barbary sheep in goats. The study also provides evidence of the virus transmission (goat-to-goat). Clinical disease was successfully reproduced, and the study provided critical information about the diagnostic tools to be used and shed some light on molecular epidemiology.

Overall the manuscript is well written, and the methodology seems appropriate.

Minor:

1- Some light editing would improve the manuscript quality. 

2- I would suggest using the term "clinical score" rather than "clinical reaction score". Or "clinical signs score". 

3- Legend of figure 3 could repeat the text's info and fully describe what is in each figure. The use of arrows may help guide the readers in some figures.

4- Legend of figure 4 could also use some extra description. 

5- For some cultures, the word  "sacrifice" is religious-related. The use of terms like euthanasia or euthanized would be more scientifically appropriate.
